# Outcomes of Transcatheter Aortic Valve Implantation Comparing Medtronic’s Evolut PRO and Evolut R: A Systematic Review and Meta-Analysis of Observational Studies

**DOI:** 10.3390/ijerph20043439

**Published:** 2023-02-15

**Authors:** Mirosław Gozdek, Łukasz Kuźma, Emil Julian Dąbrowski, Michał Janiak, Martyna Pietrzak, Karolina Skonieczna, Mikołaj Woźnica, Lidia Wydeheft, Maged Makhoul, Matteo Matteucci, Radosław Litwinowicz, Adam Kowalówka, Wojciech Wańha, Michał Pasierski, Daniele Ronco, Giulio Massimi, Federica Jiritano, Dario Fina, Gennaro Martucci, Giuseppe Maria Raffa, Piotr Suwalski, Roberto Lorusso, Paolo Meani, Mariusz Kowalewski

**Affiliations:** 1Department of Cardiology, Hospital of the Ministry of Interior, 72-122 Bydgoszcz, Poland; 2Thoracic Research Centre, Collegium Medicum Nicolaus Copernicus University, Innovative Medical Forum, 85-094 Bydgoszcz, Poland; 3Department of Invasive Cardiology, Medical University of Bialystok, 15-540 Bialystok, Poland; 4Department of Cardiac Surgery, Harefield Hospital, London UB9 6JH, UK; 5Cardiac Surgery Unit, ASST dei Sette Laghi, Department of Medicine and Surgery, University of Insubria, 21100 Varese, Italy; 6Cardiothoracic Surgery Department, Heart and Vascular Centre, Maastricht University Medical Centre (MUMC), The Cardiovascular Research Institute Maastricht (CARIM), 6229 ER Maastricht, The Netherlands; 7Department of Cardiac Surgery, Regional Specialist Hospital, 86-300 Grudziądz, Poland; 8Department of Cardiac Surgery, Upper-Silesian Heart Center, 40-752 Katowice, Poland; 9Department of Cardiac Surgery, Faculty of Medical Sciences, Medical University of Silesia, 40-752 Katowice, Poland; 10Department of Cardiac Surgery, School of Medicine in Katowice, Medical University of Silesia, 40-752 Katowice, Poland; 11Department of Cardiac Surgery and Transplantology, Central Clinical Hospital of the Ministry of Interior, Centre of Postgraduate Medical Education, 02-507 Warsaw, Poland; 12Department of Experimental and Clinical Medicine, Magna Graecia University, 88100 Catanzaro, Italy; 13Department of Cardiothoracic, Vascular Anesthesia and Intensive Care Unit, Istituto di Ricovero e Cura a Carattere Scientifico (IRCCS) Policlinico, San Donato Milanese, 20097 Milan, Italy; 14Department of Anesthesia and Intensive Care, Istituto Mediterraneo per i Trapianti e Terapie ad Alta Specializzazione (IRCCS-ISMETT), 90100 Palermo, Italy; 15Department for the Treatment and Study of Cardiothoracic Diseases and Cardiothoracic Transplantation, Istituto di Ricovero e Cura a Carattere Scientifico-Istituto Mediterraneo per i Trapianti e Terapie ad Alta Specializzazione (IRCCS-ISMETT), 90100 Palermo, Italy

**Keywords:** TAVI, transcatheter intervention, aortic stenosis, heart failure, permanent pacemaker, paravalvular leak, Evolut PRO, Evolut R, meta-analysis

## Abstract

Background: Transcatheter aortic valve implantation (TAVI) has become a broadly acceptable alternative to AV surgery in patients with aortic stenosis (AS). New valve designs are becoming available to address the shortcomings of their predecessors and improve clinical outcomes. Methods: A systematic review and meta-analysis was carried out to compare Medtronic’s Evolut PRO, a new valve, with the previous Evolut R design. Procedural, functional and clinical endpoints according to the VARC-2 criteria were assessed. Results: Eleven observational studies involving N = 12,363 patients were included. Evolut PRO patients differed regarding age (*p* < 0.001), sex (*p* < 0.001) and STS–PROM estimated risk. There was no difference between the two devices in terms of TAVI-related early complications and clinical endpoints. A 35% reduction of the risk of moderate-to-severe paravalvular leak (PVL) favoring the Evolut PRO was observed (RR 0.66, 95%CI, [0.52, 0.86] *p* = 0.002; *I*^2^ = 0%). Similarly, Evolut PRO-treated patients demonstrated a reduction of over 35% in the risk of serious bleeding as compared with the Evolut R (RR 0.63, 95%CI, [0.41, 0.96]; *p* = 0.03; *I*^2^ = 39%), without differences in major vascular complications. Conclusions: The evidence shows good short-term outcomes of both the Evolut PRO and Evolut R prostheses, with no differences in clinical and procedural endpoints. The Evolut PRO was associated with a lower rate of moderate-to-severe PVL and major bleeding.

## 1. Introduction

Transcatheter aortic valve implantation (TAVI) is considered as an alternative treatment option to surgery and is recommended not only in inoperable high-risk patients [1,2,3,4] but also in intermediate- and even low-risk individuals presenting with severe aortic valve stenosis [5,6,7,8,9,10].

Two types of transcatheter devices were at physicians’ disposal within a few years of the performing of the first procedure in 2002: the balloon-expandable Sapien**^®^** (Edwards Lifesciences, Irvine, CA, USA) and the self-expanding CoreValve**^®^** (Medtronic Incorporation, Minneapolis, MN, USA). Those early-generation transcatheter valves, despite providing good early clinical outcomes, were not without shortcomings: the relatively high rate of paravalvular leak (PVL), which was associated with increased late mortality and more adverse clinical outcomes than surgery [11,12], was of the greatest concern. To overcome these issues, successive modifications of above-mentioned devices have emerged over the past decade. Other companies have also launched TAVI sets. Introduced in March 2017, Evolut**^®^** PRO is a third iteration of Medtronic’s transcatheter bioprosthesis. It was built on Evolut R’s base and, therefore, maintains all the properties of its precursor (self-expandability as well as recapturability and resheathability for repositioning). The only difference is the addition of an external pericardial wrap to the lower part of the nitinol frame to improve sealing. The modification, unfortunately, translated into a larger size of introducer sheath dedicated to the prosthesis.

The objective of the present investigation was to evaluate and compare the short-term results (up to 30 days) of TAVI with the Evolut PRO and Evolut R in patients presenting with symptomatic severe aortic valve stenosis, with particular emphasis on PVL and major vascular complications (MVC), including serious bleeding.

## 2. Materials and Methods

### 2.1. Data Sources and Search Strategy

The systematic review and meta-analysis were performed in accordance with the MOOSE statement and PRISMA guidelines [13,14]. The MOOSE/PRISMA checklist is available in Appendix A. We searched PubMed, Google Scholar, ClinicalKey and the Web of Science until November 2022. The search terms were Evolut PRO, Evolut R, Evolut PRO vs. Evolut R, Evolut R vs. Evolut PRO and transcatheter valve or transcatheter aortic valve. The literature was limited to peer-reviewed articles published in English. The references of the original articles were reviewed manually and cross-checked.

### 2.2. Selection Criteria and Quality Assessment

Studies were included if they met both the following criteria: (1) they were human studies; and (2) they were study or study arms directly comparing strategies for transcatheter aortic valve replacement with the Evolut R and Evolut PRO. Studies were excluded if: (1) they were in-vitro studies; (2) they were single arm studies; or (3) the outcomes of interest were not reported. No restrictions regarding the number of patients included or the characteristics of the population were imposed. Two reviewers (MG and MK) selected the studies for inclusion, and extracted the studies and patients’ characteristics of interest and relevant outcomes. Two authors (MG and MK) independently assessed the trials’ eligibility and risk of bias. Any divergences were resolved by consensus. The quality of the studies was appraised with ROBINS-I (Risk of Bias in Non-randomised Studies - of Interventions), a tool used for the assessment of bias (the selection of the study groups; the comparability of the groups; and the ascertainment of either the exposure or outcome of interest) in cohort studies included in a systematic review and/or meta-analysis [15].

### 2.3. Endpoints Selection

Endpoints were established according to the Valve Academic Research Consortium-2 (VARC-2) definitions [16]. A procedural outcome of interest was needed for more than one prosthesis utilization and other TAVI-related complications (pooled together: conversion to surgery, coronary obstruction, ventricular septal perforation, mitral valve apparatus damage/dysfunction, endocarditis, cardiac tamponade, prosthetic valve thrombosis or malpositioning–migration, embolization or ectopic deployment). The clinical endpoints assessed included serious bleeding (life-threatening and/or major), major vascular complications (MVC), cerebrovascular accident (CVA) (stroke and/or TIA), peri-procedural myocardial infarction, permanent pacemaker implantation (PPI) and 30-day mortality. The functional outcomes were moderate-to-severe paravalvular leak (PVL), mild PVL, mean transprosthetic gradient and prosthesis–patient mismatch (PPM).

### 2.4. Statistical Analysis

Data were analyzed according to the intention-to-treat principle wherever applicable. Risk Ratios (RR) and 95% Confidence Intervals (95%CI) served as primary index statistics for dichotomous outcomes. For continuous outcomes, Mean Difference (MD) and corresponding 95% CI were calculated using the random-effects model. To overcome the low statistical power of the Cochran Q test, the statistical inconsistency test *I*^2^ = [(Q_df)/Q] × 100%, where Q is the chi-square statistic and df its degrees of freedom, was used to assess heterogeneity [17]. This examines the percentage of inter-study variation, with values ranging from 0% to 100%. An *I*^2^ value of less than 40% indicates no obvious heterogeneity, values between 40%–70% are suggestive of moderate heterogeneity and *I*^2^ more than 70% is considered as high heterogeneity. Because of the high degree of heterogeneity anticipated among predominantly non-randomized trials, an inverse variance (DerSimonian-Laird) random-effects model was applied as a more conservative approach for observational data accounting for between- and within-study variability. Whenever a single study reported median values and interquartile ranges instead of mean and standard deviation (SD), the latter were approximated as described by Wan and colleagues [18]. If there were “0 events” reported in both arms, the calculations were repeated, as a sensitivity analysis, using Risk Difference (RD) and respective 95% CI. Review Manager 5.4.1 (The Cochrane Collaboration, 2020) was used for the statistical computations. Finally, ***p***-values ≤ 0.05 were considered statistically significant and reported as two-sided, without adjustment for multiple comparisons.

## 3. Results

### 3.1. Study Selection

The study selection process and the reasons for the exclusion of some studies are described in Figure 1. A systematic search of the online databases allowed the collection of 762 potentially eligible records that were retrieved for scrutiny. Of these, 751 were excluded because they were not pertinent to the design of the meta-analysis or did not meet the explicit inclusion criteria. Eleven observational studies [19,20,21,22,23,24,25,26,27,28,29] (among them seven multi-center registries), including N = 12,363 patients, were eventually included in the analysis. Potential sources of study bias were analyzed with the use of the components recommended by the ROBINS-I tool, and the results are shown in Appendix A. Overall, the retrospective studies showed a moderate risk of bias. The most common biases arose from participants’ selection for the study by designated heart teams and the subjective distribution of the participants within the study arms by designated operators.

A summary of the valve characteristics is shown in Table 1. The Evolut PRO was built on the Evolut R platform. The only two differences between the valves are the external pericardial wrap and the wider introducer sheath in the PRO platform. The studies’ baseline characteristics are reported in Table 2. Appendix A lists the selection criteria for the procedure and valve, as well as the inclusion and exclusion criteria within particular studies. The study of Dallan et al. [20] concerns TAVI for degenerated surgical bioprostheses. In the study of Modolo et al. [26], the data on PVL come from an analysis of aortograms.

### 3.2. Patients’ Characteristics

The groups treated with the Evolut PRO (N = 3439) and Evolut R (N = 8924) substantially differed regarding patients’ age (*p* < 0.001), sex (*p* < 0.001) and STS–PROM risk profile (*p* < 0.001). Patients from the Evolut PRO arm were older (80.13 ± 7.65 vs. 77.58 ± 8.79) and were more often female (62.92% vs. 54.71%) but had a lower risk profile (6.79 ± 6.4 vs. 7.34 ± 5.6). The groups did not differ regarding BMI (*p* = 0.732) or NYHA III/IV status (*p* = 0.558). Transfemoral TAVI was performed in 95.5% of the Evolut PRO group and 94.1% of the Evolut R arm.

Aortic valve baseline parameters, such as native annulus diameter and mean transaortic gradient, were comparable between the groups (*p* = 0.121 and *p* = 0.709, respectively).

The effective orifice area was larger in the Evolut PRO recipients (0.70 ± 0.21 vs. 0.68 ± 0,17 for the Evolut PRO and Evolut R, respectively, *p* = 0.002). Patients treated with the Evolut PRO received smaller prostheses; the mean size of the implanted valve was 26.57 ± 0.86 in the Evolut PRO and 27.96 ± 1.59 in the Evolut R group (*p* < 0.001). Patients’ baseline and detailed procedural characteristics are available in Table 3 and Appendix A.

### 3.3. Procedural Outcomes

The need for the utilization of more than one prosthesis during initial implantation was low in both the Evolut PRO (0.87%, 26 of 2973 cases) and Evolut R groups (1.18%, 94 of 7972 cases), with statistical significance in favor of the Evolut PRO (RR 0.52, 95%CI, [0.30, 0.89] *p* = 0.02; I2 = 8%) (Appendix A). There was no difference in the pooled estimate of the other TAVI-related complications between the groups: 0.45% (13 of 2900 cases) vs. 0.49% (37 of 7526 cases) for the Evolut PRO and Evolut R, respectively (RR 0.71, 95%CI, [0.22, 2.32] *p* = 0.57; I^2^ = 60%) (Appendix A).

### 3.4. Functional Outcomes

Eleven (N = 12,363) and seven (N = 10,862) studies were included in the analysis of moderate-to-severe and mild PVL, respectively. A reduction of almost 35% in the risk of moderate-to-severe PVL favoring the Evolut PRO was observed (RR 0.66, 95%CI, [0.52, 0.86] *p* = 0.002; I^2^ = 0%), with corresponding event rates of 2.41% (83 of 3439) and 3.03% (270 of 8924) for the Evolut PRO and Evolut R, respectively (Figure 2A). No difference regarding mild PVL was noticed between the devices (RR 0.94, 95%CI, [0.86, 1.02] *p* = 0.16; I^2^ = 0%) (Figure 2B).

Mean transprosthetic gradient was similar in the Evolut PRO and Evolut R arm (MD −0.11 95%CI, [−0.78; 0.56] mmHg; *p* = 0.74) (Appendix A). Moreover, there was no difference between the Evolut PRO and Evolut R in term of at-least-moderate prosthesis–patient mismatch (RR 1.05, 95%CI, [0.97, 1.14] *p* = 0.25; I^2^ = 0%) (Appendix A).

### 3.5. Clinical Outcomes

Nine studies, enrolling 11,159 and 10,476 patients, respectively, provided data for the analysis of serious (life-threatening and major) bleeding and major vascular complications (MVC). The Evolut PRO-treated patients demonstrated a reduction of over 35% in the risk of serious bleeding as compared with the Evolut R group (RR 0.63, 95%CI, [0.41, 0.96]; *p* = 0.03; I^2^ = 39%), with corresponding frequencies of 2.63% (76 of 2901) vs. 5.02% (406 of 8083), respectively (Figure 3A). No differences regarding MVC were seen between the two devices (RR 0.77, 95%CI, [0.54, 1.08] *p* = 0.13; I^2^ = 7%) (Figure 3B).

In addition, the risk of other clinical endpoints was no different between the devices, as seen in the results for 30-day mortality (RR 0.93, 95%CI, [0.69, 1.25] *p* = 0.63; I^2^ = 0%), peri-procedural MI (RR 1.31, 95%CI, [0.42, 4.05] *p* = 0.64; I^2^ = 40%) and CVA (RR 0.81, 95%CI, [0.61, 1.08] *p* = 0.15; I^2^ = 0%), (Appendix A).

Data on PPI were available in nine studies including 11,149 patients. The risk of PPI was numerically higher in the Evolut PRO group than in the Evolut R group, with corresponding rates of 11.12% (341 of 3066) and 8.96% (724 of 8083), but was without statistical significance (RR 0.82, 95%CI, [0.66, 1.03] *p* = 0.09; I^2^ = 50%) (Figure 4). Over 60% of PPIs were reported in a single study [21] and a big study effect cannot be excluded.

### 3.6. Sensitivity Analysis

The exclusion of individual studies one at a time and repeating the calculations for moderate-to-severe PVL and major bleeding did not change the results; similarly, applying RD analysis for studies reporting “0” events did not alter the estimates (Appendix A).

## 4. Discussion

Our study is the first meta-analysis to compare procedural, clinical and functional results between Medtronic’s Evolut PRO and Evolut R. By pooling data from 11 observational studies, we could demonstrate good outcomes regarding the short-term performance of both devices.

The main finding of our investigation is that the Evolut PRO was associated with a significantly lower rate of paravalvular leaks and a trend towards a lower rate of life-threatening or major bleeding. Other functional as well as clinical and procedural outcomes did not differ between the devices. A substantial imbalance between the groups regarding age, sex and risk profile was noticed across the studies. The Evolut PRO population was older and more often of female dominance but had a lower STS-risk score.

Moderate to severe PVL has been associated with a higher long-term mortality rate [30]. Other long-term follow-up data have suggested that even mild PVL negatively affects late mortality after implantation of the self-expanding CoreValve, an early-generation prosthesis [12,31].

Ando et al. [32], in a meta-analysis including 21,018 patients, demonstrated higher all-cause mortality in patients with mild PVL compared to none/trivial PVL (RR 1.26, [1.11–1.43], *p* < 0.001]. Medtronic’s intention was to decrease the relatively high rate of PVL of the Evolut R prosthesis. The same platform was used to build the new-iteration Evolut Pro. An external pericardial skirt was added to the lower part of the frame to improve sealing and obtained a reduction of 35% in PVL compared to its predecessor in our analysis.

Because the pericardial wrap had been added onto the lower part of the Evolut PRO frame, the introducer sheath had to be increased in size from 14F to 16F; therefore, an increased rate of vascular complications and/or bleeding was anticipated. Barbanti et al. [33] showed that the lower-profile sheath was associated with a lower incidence of major vascular complications (0.5 vs. 10.5%, *p*< 0.001) and a lower rate of life-threatening or major bleeding (3.4 vs. 8.3%, *p* = 0.038). Despite the larger diameter of the introducer, we observed no difference in major vascular complications between the Evolut PRO and Evolut R group, and an even lower rate of life-threatening or major bleeding in the Evolut PRO recipients. While we could not address these issues directly, since we were limited by the availability of data reported in the individual studies included, one of the possible explanations may be the more frequent use of percutaneous closure devices in the group of patients treated with the larger sheaths. Moreover, the pre-procedural analysis of the computed tomography angiograms and careful assessment of the potential anatomical bleeding risk factors may have influenced the use of Evolut R systems in the ambiguous cases. Lastly, multidisciplinary TAVI teams’ growing experience appears to be another plausible explanation for this observation. All these factors could have resulted in the appropriate selection of patients suitable for the Evolut PRO system and, as a consequence, in lower bleeding rates associated with the procedure, and are of undisputable importance to the interventional community [34].

Interestingly, the early Medtronic Evolut PRO US Clinical Study that investigated outcomes of the new aortic valve system reported that, despite the addition of the outer pericardial wrap, PPI rates were even lower than in the previous generation (11.8% vs. 19.7%, respectively) [35]. This observation was partially explained by increased operator experience and comfort, resulting in a lower mean implantation depth. Although the study was relatively small and the results should have been considered only hypothesis-generating, it raised hopes that the rate of serious conduction disturbances after transcatheter interventions will decrease. However, our analysis showed the opposite—PPI was numerically more common in the Evolut PRO group than in the Evolut R group. The PPI rate in the Evolut PRO group was similar to that reported in Forrest et al.’s clinical study, whereas that in the Evolut R was unexpectedly lower than in previous reports [7,35,36]. As suggested above, the low PPI rate in the older-generation aortic valve system group may be related to the TAVI teams’ growing experience. This is an important finding, suggesting that for skilled operators the rate of serious conduction disturbances is comparable with those reported in the surgical arms of randomized clinical trials [7].

Several limitations need to be acknowledged. Firstly, the current investigation consists only of observational studies; therefore, selection bias could arise, particularly in associated with the time frames imposed (e.g., the Evolut PRO was available after the Evolut R). With the experience gained over the years during which the Evolut R was being implanted, some complications may have been avoided in the Evolut PRO generation. Several studies did not report on the outcomes of interest, which makes the conclusions regarding these outcomes valid only to the extent that the remaining studies allow. Secondly, the studies comparing the Evolut PRO and Evolut R have thus far reported only short-term outcomes; data regarding long-term mortality and re-interventions, and in particular how these risks are affected by the initial presence of PVL, are of great interest. Finally, the quality and the risk of bias in the above observational studies were assessed as moderate in most of the studies; however, the randomization of patients to a current design valve and previous generation prosthesis could generate ethical questions. Moreover, publication bias and big study effect cannot be totally excluded, and indeed, two of the included studies [20,21] constituted over 70% of the included population, and therefore these studies’ individual limitations may also reflect the limitations of the current analysis. However, sensitivity analyses, in which each individual study was successively excluded and the calculation repeated in its absence, changed neither the direction nor the magnitude of the estimates.

## 5. Conclusions

The evidence shows good short-term outcomes of both the Evolut PRO and Evolut R prostheses, with no differences in the clinical and procedural endpoints. Implantation of the Evolut PRO was associated with a statistically significantly lower rate of moderate-to-severe PVL. These benefits might, in consequence, further translate into improved long-term clinical outcomes.

## Figures and Tables

**Figure 1 ijerph-20-03439-f001:**
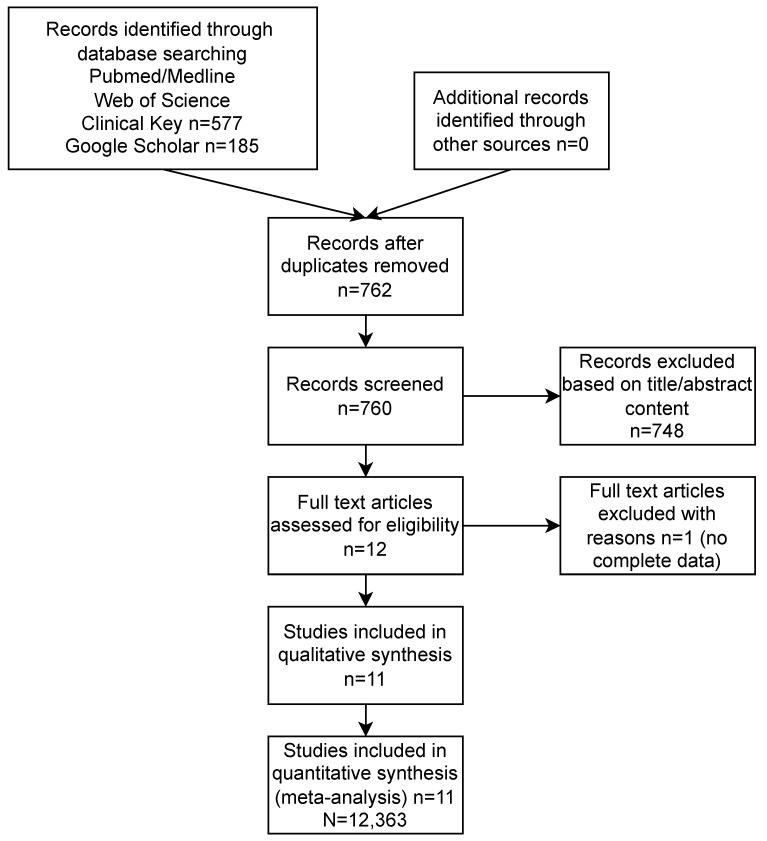
Study flow chart along with reasons for study exclusion.

**Figure 2 ijerph-20-03439-f002:**
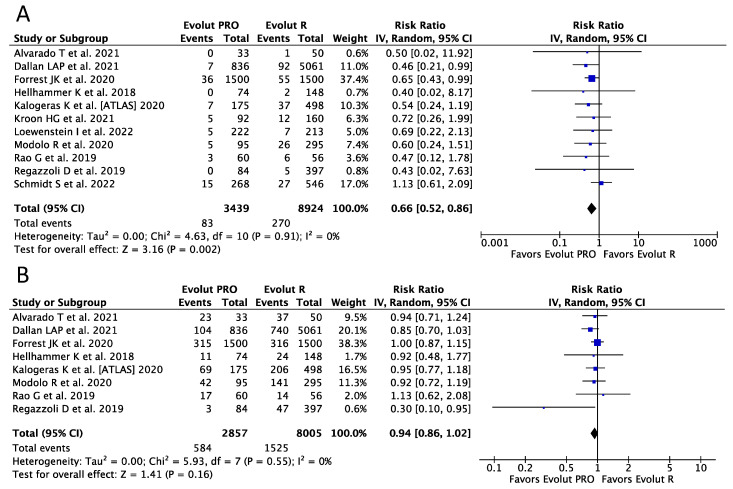
Risk Ratios (RRs) and corresponding 95% Confidence Intervals (CIs) for the comparison of Evolut PRO and Evolut R devices [19,20,21,22,23,24,25,26,27,28,29]; (**A**) moderate-to-severe paravalvular leak; (**B**) mild paravalvular leak. Each square represents study point estimate; diamonds reflect the overall effect. IV, inverse variance.

**Figure 3 ijerph-20-03439-f003:**
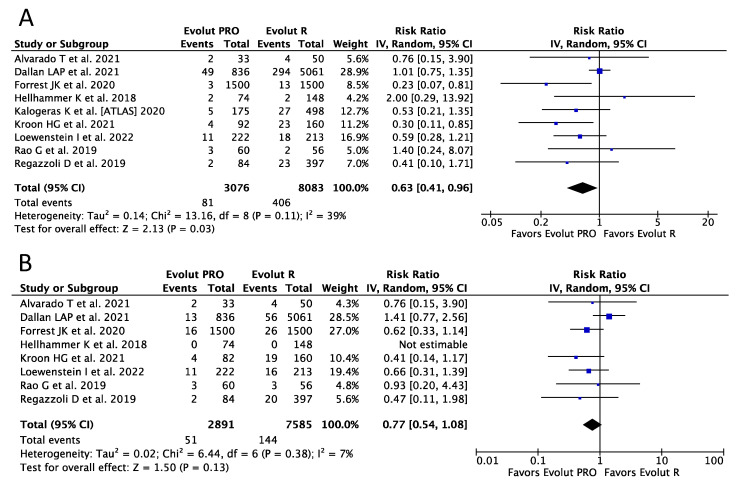
Risk Ratios (RRs) and corresponding 95% Confidence Intervals (CIs) for the comparison of Evolut PRO and Evolut R devices [19,20,21,22,23,24,25,27,28]; (**A**) serious bleeding; (**B**) major vascular complications. Each square represents study point estimate; diamonds reflect the overall effect. IV, inverse variance.

**Figure 4 ijerph-20-03439-f004:**
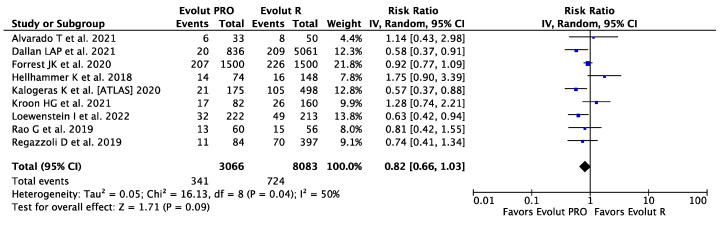
Risk Ratios (RRs) and corresponding 95% Confidence Intervals (CIs) for the comparison of Evolut PRO and Evolut R devices in terms of permanent pacemaker implantation [19,20,21,22,23,24,25,27,28]. Each square represents study point estimate; diamonds reflect the overall effect. IV, inverse variance.

**Table 1 ijerph-20-03439-t001:** Summary of the valve characteristics.

Evolut PRO	Evolut R
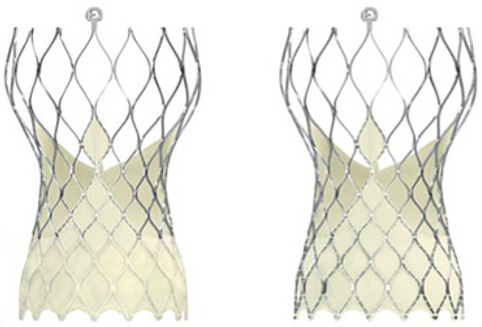	
Supra-annular design Self-expanding nitinol frame Strong and pliable porcine tissue Ability to recapture and resheath the prosthesis 16-F introducer sheath Valve diameter: 23 mm, 26 mm, 29 mm Unique valve design with an outer wrap	Supra-annular design Self-expanding nitinol frame Strong and pliable porcine tissue Ability to recapture and resheath the prosthesis 14-F introducer sheath Valve diameter: 23 mm, 26 mm, 29 mm, 34 mm -

Evolut PRO and Evolut R devices. The two self-expandable Evolut R and Evolut PRO devices. Reproduced with modification from Kalogeras et al. [23] Copyright © 2020 Elsevier B.V.

**Table 2 ijerph-20-03439-t002:** Baseline characteristics of included studies.

Study	Design	Intervention	Cohort	Follow-Up (Months)	VARC-2 Definitions	ROBINS-I
Alvarado T et al., 2021 [19]	SC RCS	Evolut PRO	33	12.0 ± 0.2	yes	moderate
		Evolut R	50	11.0 ± 0.5		
Dallan LAP et al., 2021 [20]	MC PCS	Evolut PRO	836	12	yes	moderate
		Evolut R	5061			
Forrest JK et al., 2020 [21]	MC RCS, PM	Evolut PRO	1500	NR	yes	moderate
		Evolut R	1500			
Hellhammer K et al., 2018 [22]	SC RCS, PM	Evolut PRO	74	1	yes	moderate
		Evolut R	148			
Kalogeras K et al. (ATLAS) 2020 [23]	MC RCS	Evolut PRO	175	12	yes	moderate
		Evolut R	498			
Kroon HG et al., 2021 [24]	SC PCS	Evolut PRO	92	1	yes	moderate
		Evolut R	160			
Loewenstein I et al., 2022 [25]	MC RCS, PM	Evolut PRO	222	1	yes	moderate
		Evolut R	213			
Modolo R et al., 2020 [26]	MC RCS	Evolut PRO	95	NR	NR	moderate
		Evolut R	295			
Rao G et al., 2019 [27]	SC RCS	Evolut PRO	60	1	yes	moderate
		Evolut R	56			
Regazzoli D et al., 2019 [28]	MC RCS	Evolut PRO	84	12	yes	moderate
		Evolut R	397			
Schmidt S et al., 2022 [29]	MC RCS	Evolut PRO	268	NR	yes	moderate
		Evolut R	546			

SC, single center; MC, multicenter; RCS, retrospective cases series; PCS, prospective cohort study; PM, propensity match; VARC-2, Valve Academic Research Consortium-2; ROBINS-I, risk of bias in non-randomized studies - of intervention; NR, not reported.

**Table 3 ijerph-20-03439-t003:** Patients’ baseline and procedural characteristics.

Study	Intervention	Age	Females (%)	BMI (kg/m2)	NYHA III/IV(%)	STS–PROM (%)	EuroSCORE II (%)	Mean Gradient [mmHg]	Annulus diameter (mm)	Transfemoral Access (%)
Alvarado T et al., 2021 [19]	Evolut PRO	85.0 ± 4.0	79	NR	48	NR	NR	42.0 ± 13.0	23.0 ± 1.8	100
	Evolut R	85 ± 0.5.0	52		68			45 ± 0.18.0	24.0 ± 4.3	100
Dallan LAP et al., 2021 [20]	Evolut PRO	74.6 ± 10.3	41.6	NR	79.7	7.2 ± 7.1	NR	NR	NR	95.4
	Evolut R	75.2 ± 10.5	41.2		80.4	7.7 ± 6.5				95.0
Forrest JK et al., 2020 [21]	Evolut PRO	81.6 ± 7.7	65.9	NR	75.5	7.2 ± 4.8	NR	42.4 ± 14.4	NR	94.1
	Evolut R	81.3 ± 7.5	65.2		78.1	7.2 ± 4.5		43.3 ± 15.2		93.5
Hellhammer K et al., 2018 [22]	Evolut PRO	81.4 ± 4.5	67.5	26.2 ± 4.4	NR	NR	24.9 ± 12.5 *	NR	23.8 ± 1.8	100
	Evolut R	81.2 ± 5.6	74.3	26.9 ± 5.8			24.7 ± 13.7 *		23.0 ± 1.9	100
Kalogeras K et al. (ATLAS) 2020 [23]	Evolut PRO	82.3 ± 6.3	59.4	26.8 ± 8.4	NR	NR	9.4 ± 7.2 *	40.3 ±19.9	NR	100
	Evolut R	81.7 ± 7.2	58.8	26.8 ± 5.7			14.4 ±9.2 *	46.5 ±18.8	NR	94
Kroon HG et al., 2021 [24]	Evolut PRO	89.7 ± 2.2	54	27.0 ± 5.0	68	4.4 ± 3.1	NR	NR	23.7 ± 1.8	97
	Evolut R	78.7 ± 2.1	46	27.0 ± 5.0	58	4.4 ± 2.8			23.7 ± 1.9	89
Loewenstein I et al., 2022 [25]	Evolut PRO	81.1 ± 6.7	62.2	28.2 ± 5.2	NR	3.9 ± 2.4	4.0 ± 3.8	48.3 ± 14.8		
	Evolut R	82.2 ± 7.6	63.4	27.6 ± 5.1		5.0 ± 4.0	5.6 ± 6.1	47.4 ± 17.6		
Modolo R et al., 2020 [26]	Evolut PRO	NR	NR	NR	NR	NR	NR	NR	NR	NR
	Evolut R									
Rao G et al., 2019 [27]	Evolut PRO	84.6 ± 6.3	82	27.5 ± 6.8	NR	7.1 ± 4.0	NR	NR	NR	95.1
	Evolut R	80.4 ± 9.4	53.6	28.8 ± 7.2		6.5 ± 4.8				94.6
Regazzoli D et al., 2019 [28]	Evolut PRO	83.5 ± 0.7	89.3	26.5 ± 0.9	81	5.5 ± 0.4	NR	51.6 ± 1.9	21.4 ± 0.1	91.7
	Evolut R	82.0 ± 0.4	89.2	26.8 ± 0.4	74.1	5.9 ± 0.3		50.6 ± 0.8	21.2 ± 0.1	92.4
Schmidt S et al., 2022 [29]	Evolut PRO	82.0 ± 5.6	61.2	26.5 ± 5.0	NR	NR	4.7 ± 4.3	48.1 ± 19.5	23.4 ± 1.7	NR
	Evolut R	82.1 ± 6.1	61.2	26.8 ± 5.2			5.7 ± 6.8	34.9 ± 20.7	23.8 ± 2.5	

BMI, body mass index; NYHA, New York Heart Association; STS–PROM, Society of Thoracic Surgeons Predicted Risk of Mortality; EuroSCORE, European System for Cardiac Operative Risk Evaluation; NR, not reported. * logistic EuroSCORE.

## Data Availability

Data available from corresponding author upon reasonable request.

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
