# Peer review of "Outcomes of Transcatheter Aortic Valve Implantation Comparing Medtronic’s Evolut PRO and Evolut R: A Systematic Review and Meta-Analysis of Observational Studies"

_ijerph, 2023, doi:10.3390/ijerph20043439_

Round 1
Reviewer 1 Report
The authors performed a meta-analysis of the usage of Evolution Pro and Evolution R stents from Medtronics. The overall study is useful for the medical community. I will suggest the authors put all the data they analyzed in an excel sheet and put it in an online repository. Some english spell check is required.
Also write a paragraph and a table explaining the key differences between Evolution Pro and Evolution R to show why they are different in terms of structure, function, cost, ease of administration in the body etc.
After all these are addressed, it can be published.
Author Response
Comment 1: The authors performed a meta-analysis of the usage of Evolution Pro and Evolution R stents from Medtronics. The overall study is useful for the medical community. I will suggest the authors put all the data they analyzed in an excel sheet and put it in an online repository. Some english spell check is required.
Reply 1: Thank you for the comment; please find the excel with our source data attached; we are not aware of any other MDPI repository; English has been checked and minor corrections applied
Changes 1: throughout the MS
Comment 2: Also write a paragraph and a table explaining the key differences between Evolution Pro and Evolution R to show why they are different in terms of structure, function, cost, ease of administration in the body etc.
Reply 2: We are grateful for this comment. Valve characteristics are available as Table 1. In brief, Evolut PRO was built on the Evolut's R platform. The only two differences between the valves are: external pericardial wrap and wider introducer sheath in the EvPRO platform. We now state that also in the results and discussion
Changes 2: please see results (2nd paragraph) and discussion (lines 264) modified
After all these are addressed, it can be published.
Reply: Thank you.
Reviewer 2 Report
The authors carry out a systematic review and meta-analysis to compare Medtronic’s Evolut PRO, a new valve with the previous Evolut R design. Procedural, functional and clinical endpoints according to VARC-2 criteria were assessed. Eleven observational studies enrolling N=12,363 patients were included. Evolut PRO patients differed regarding age (p<0.001), sex (p<0.001), and STS-PROM estimated risk. There was no difference between the two devices in terms of TAVI-related early complications and clinical endpoints. A 35% reduction of the risk of moderate-to-severe para-valvular leak (PVL) favoring Evolut PRO was observed (RR 0.66, 95%CI, [0.52, 0.86] P=0.002; I2=0%). Similarly, Evolut PRO treated patients demonstrated over 35% reduction of the risk of serious bleedings as compared with Evolut R (RR 0.63, 95%CI, [0.41, 0.96]; P=0.03; I2=39%) without differences in major vascular complications. The authors conclude that good short-term outcomes are associated with both Evolut PRO and Evolut R prostheses with no differences in clinical and procedural endpoints. Evolut PRO was associated with lower rate of moderate-to-severe PVL and major bleeding.
1. The authors report that Evolut PRO was associated with lower rate of moderate-to-severe PVL and major bleeding. These findings are of interest to interventional community.
2. How do the authors explain the fact that Evolut PRO was associated with lower rare of major bleeding despite a larger introducer sheath?
Author Response
The authors carry out a systematic review and meta-analysis to compare Medtronic’s Evolut PRO, a new valve with the previous Evolut R design. Procedural, functional and clinical endpoints according to VARC-2 criteria were assessed. Eleven observational studies enrolling N=12,363 patients were included. Evolut PRO patients differed regarding age (p<0.001), sex (p<0.001), and STS-PROM estimated risk. There was no difference between the two devices in terms of TAVI-related early complications and clinical endpoints. A 35% reduction of the risk of moderate-to-severe para-valvular leak (PVL) favoring Evolut PRO was observed (RR 0.66, 95%CI, [0.52, 0.86] P=0.002; I2=0%). Similarly, Evolut PRO treated patients demonstrated over 35% reduction of the risk of serious bleedings as compared with Evolut R (RR 0.63, 95%CI, [0.41, 0.96]; P=0.03; I2=39%) without differences in major vascular complications. The authors conclude that good short-term outcomes are associated with both Evolut PRO and Evolut R prostheses with no differences in clinical and procedural endpoints. Evolut PRO was associated with lower rate of moderate-to-severe PVL and major bleeding.
Comment 1. The authors report that Evolut PRO was associated with lower rate of moderate-to-severe PVL and major bleeding. These findings are of interest to interventional community.
Comment 2. How do the authors explain the fact that Evolut PRO was associated with lower rare of major bleeding despite a larger introducer sheath?
Reply 1 and 2: Thank you for this comment; one of the possible explanations may include the more eager use of percutaneous closure devices in the group of patients treated with the larger sheaths. Moreover, the pre-procedural analysis of the computed tomography angiograms and the careful assessment of potential anatomical bleeding risk factors may have influenced the use of Evolut R systems in the ambiguous cases. Lastly, multidisciplinary TAVI teams growing experience appear to be another plausible explanation of that observation. All of these factors could have resulted in the appropriate selection of patients suitable for the Evolut PRO system and as a consequence in lower bleeding rates associated with the procedure. Unfortunately, we could not address these since limited to the data availability from single studies reporting.
Changes 1 and 2: please see the 5th paragraph of the discussion (lines 264-282) modified
Reviewer 3 Report
Please see the attachment.

Author Response
Comment 1
Even when conducting meta-analyses of randomized clinical trials (RCTs), systematic errors
are noted. As stated by Barili et al. (1) "... the RCT design does not protect from biases other than nonrandom allocation. In RCTs comparing TAVI vs SAVR, there were systematic
imbalances in the proportion of DAT, loss to follow-up, and receipt of additional procedures and additional myocardial revascularization that can pose a serious threat to internal validity due to high risk of performance and attrition biases". The peer-reviewed meta-analysis included retrospective studies, which are even more likely to be biased than RCTs. We would like the authors to justify the need for a meta-analysis with an initially large number of limitations.
Reply 1
Thank you for this insightful commnent. The rationale behind the current analysis is that we decided to perform a meta-analysis of observational studies driven by the of lack of randomized ones. Constant and rapid development of TAVI techniques and devices is the leading cause of that. Evolut PRO was developed to overcome the shortcomings of its predecessor - Evolut R, especially paravalvular leakage (by added external pericardial skirt). Performing randomized trial assessing current and previous generation prosthesis could generate ethical questions for sure.
We agree on the presence of biases inherent to studies that are observational in nature; on the other hand these represent the real-world scenarios; we were able to demonstrate a 35% lower rate of moderate-to-severe PVL with Evolut PRO and now we expect the data to become available on how this phenomenon affects long-term mortality and re-interventions which may be of even greater interest and use to the medical community.
Changes 1: We have now added a respective paragraph in the limitations section; please see lines 299-310
Objection 2.
The largest number of patients (72%) in the meta-analysis was represented by data from only two studies (2.3). Therefore, when considering the limitations of a meta-analysis as a whole, the limitations of these individual studies must also be taken into account. So, in a study by Dallan et al (2), when comparing Evolut PRO and R valves, we did not statistically adjust for differences in baseline performance between groups or valve sizes. Also, this study did not have access to information on balloon valve fractures or information on how many patients were denied TAV-in-SAV due to complex anatomy and who were denied the procedure. In the study by Forrest et al (3), in turn, the proportion of women was higher than in the general TAVR population, and differences in early outcomes and complications depend on gender (etc). Therefore, the study limitation section should be expanded.
Reply 2: We are grateful for this remark, we agree with the comment and have now expanded the limitations section accordingly.
Changes 2: We have now added a respective paragraph in the limitations section; please see lines 310-316
Comment 3.
In addition, I was somewhat surprised by the number of authors of this meta-analysis.Moreover, most of these co-authors participated only in writing the text of the manuscript(according to the Author Contributions section). Was the contribution of all these respected scientists really so great that it deserves co-authorship in the article?
Reply 3: Thank you for this comment and i am happy to clarify. Authors Michał Janiak, Martyna Pietrzak, Karolina Skonieczna, Mikołaj Woźnica, Lidia Wydeheft are medical students and they are only at the beginning of their scientific path. They are just learning to perform researches and write articles being recently introduced to the Thoracic Research Centre (trc.org.pl). It is an international group of established researchers in the field of cardiovascular disease that toghether are promoting research and other scientific activities in the young doctors. While indeed, their "written contribution" was limited to writing the text of the manuscript, they participated in all the stages of creating the current publication with a huge engagement and enthusiasm, accordingly to their skills.
Changes 3. none
Round 2
Reviewer 3 Report
I am satisfied with the answers of the authors to my questions, I have no other comments.